# Research on Explosive Hardening of Titanium Grade 2

**DOI:** 10.3390/ma16020847

**Published:** 2023-01-15

**Authors:** Michał Gloc, Sylwia Przybysz-Gloc, Marcin Wachowski, Robert Kosturek, Rafał Lewczuk, Ireneusz Szachogłuchowicz, Paulina Paziewska, Andrzej Maranda, Łukasz Ciupiński

**Affiliations:** 1Faculty of Materials Science and Engineering, Warsaw University of Technology, 141 Woloska St., 02-507 Warsaw, Poland; 2Institute of High Pressure Physics, Polish Academy of Sciences (Unipress), 29/37 Sokolowska St., 01-142 Warsaw, Poland; 3Faculty of Mechanical Engineering, Military University of Technology, 2 gen. S. Kaliskiego St., 00-908 Warsaw, Poland; 4Łukasiewicz Research Network—Institute of Industrial Organic Chemistry, 6 Annopol St., 03-236 Warsaw, Poland

**Keywords:** titanium, explosive hardening, corrosion, microstructure

## Abstract

In this investigation, three different explosive materials have been used to improve the properties of titanium grade 2: ammonal, emulsion explosives, and plastic-bonded explosives. In order to establish the influence of explosive hardening on the properties of the treated alloys, tests were conducted, including microhardness testing, microstructure analysis, and tensile and corrosion tests. It has been found that it is possible to achieve a 40% increase in tensile strength using a plastic explosive (PBX) as an explosive material. On the other hand, the impact of the shock wave slightly decreased the corrosion resistance of titanium grade 2. The change in corrosion rate is less than 0.1µm/year, which does not significantly affect the overall corrosion resistance of the material. The reduction in corrosion resistance is probably due to the surface geometry changes as a result of explosive treatment.

## 1. Introduction

The development of light metal technology creates new perspectives for many industries, including aviation, automotive, and armaments. Among the light alloys, which are one of the main drivers of the progress of modern technology, special attention should be paid to titanium alloys, which are used not only in aircraft structures but also, due to their biocompatibility, in medicine [1,2]. The titanium alloy commonly used in implantology is Ti6Al4V (also known as grade 5 titanium), a two-phase alloy with high specific strength, corrosion resistance, and excellent fatigue properties [3,4,5]. Although this material is commonly used for, among other things, hip prostheses, the relatively high content of aluminum may lead to a decrease in the biocompatibility of the implant [6]. For this reason, various solutions are being sought to reduce the share of aluminum, e.g., by developing new alloys (e.g., Ti5Al4V); however, they are characterized by lower strength parameters [7]. Another potential solution is the use of technically pure titanium (e.g., grade 1, grade 2, and grade 11) and subjecting it to appropriate treatment to give it the highest possible strength parameters, similar to the strength of the Ti6Al4V alloy [8,9]. Such a significant strengthening can be obtained by severe plastic deformation, allowing for a strong fragmentation of the grain structure of the material and obtaining a high degree of grain boundary strengthening [10]. However, severe plastic deformation, necessary to obtain a fine-grained structure, is usually associated with significant geometric changes to semi-finished products and requires a series of post-processing operations to give the product the desired form [11,12]. Hence, another potential approach is to strengthen the titanium alloy by strongly damaging its structure through the action of a shock wave, which can be introduced into the material in the process of explosive strengthening [13].

Explosive metal processing has a number of advantages over classic metal strengthening techniques by plastic deformation, and the most important include short process time and relatively small changes in the thickness of the workpieces in relation to the obtained hardening degree [14,15,16]. In this process, the loading of the metal element can be realized both by the direct action of the detonation products and by using them to drive the intermediate plate, which, hitting the workpiece, generates a pressure impulse [15]. It is important that after reaching the dynamic limit of elasticity of a given workpiece, two waves are created: elastic and plastic. The increase in dynamic stress increases the speed of the plastic wave, which allows it to catch up with the elastic wave, creating a shock wave [17]. The resulting shock wave causes a strong defect in the structure of the workpiece, including an increase in dislocation density and deformation twins [14,17]. The most important factor determining the degree of hardening is the amount of pressure applied. In the case of titanium alloys, a dynamic pressure load of 8 GPa causes a phase transition α → ω, i.e., a reconstruction of the hexagonal lattice into a spatially centered cubic lattice. At this pressure value and higher, the formation of a coniferous structure is observed, and the strengthening itself has its source in strongly defective phase transformation products [18,19]. Below the pressure value of 8 GPa, the hardening obtained by explosive treatment will be mainly the result of the twinning of the microstructure [19]. In both cases, a significant increase in the strength parameters of the treated titanium alloy is observed [16,19]. In the rather modest literature dealing with the impact of shock wave loading on the strength of titanium alloys, the emphasis is mainly on the microstructural aspects and microhardness distributions; thus, it does not provide accurate strength parameters for estimating the ability of the treated materials to carry operating loads.

The aim of this work was to check the possibility of increasing the basic strength parameters of a grade 2 titanium alloy by explosive treatment with the use of various explosives and blasting systems.

## 2. Materials and Methods

### 2.1. Tested Materials

The tested material was a grade 2 titanium alloy. The samples were taken from the material in the form of a sheet with dimensions of 350 × 350 × 8 mm (Wolften). The material was tested in the initial state (after rolling) and after explosive hardening with the use of various explosives: ammonal, emulsion materials, plastic explosives. The material was tested in the state after strengthening only with an explosive or an additional technological spacer in the form of a 1 mm thick steel sheet. The characteristics of the samples are shown in Table 1. The composition of the explosives used for the strengthening process is shown in Table 2.

### 2.2. Explosive Strengthening System

An important factor when strengthening materials with the explosive method is a properly prepared setup (Figure 1), which allows for the generation of an optimal shock wave. This wave interacts directly with the tested material. This is an important factor that allows the shock wave to spread evenly throughout the material. In the process of producing materials reinforced with the explosive method, it is important to properly prepare the experimental setup so that the pressure distribution is uniform over the entire surface of the sample. This allows the metal to deform evenly. The direct contact of the explosive with the sample results in the maximum transfer of the explosion energy.

The preparation of the sheets for the blast strengthening process consisted of the appropriate adjustment of the loaded material by checking the geometrical properties of the sheet, adjusting the technological system, and then subjecting it to the forces generated by the detonation wave. The first stage consists in cleaning the contact surface of the sheet by abrasive treatment and then carrying out visual examinations in order to determine possible defects on the surface of the material in the form of cracks, which could cause failure in the correct course of the deformation process. Then, the geometry of the sheets was adjusted by spark cutting. The next stage was the preparation of the technological station where the processing will be carried out. In order to locate an explosive on the plate evenly, appropriate parts were obtained by a 3D printing method from polylactide (PLA) (green elements in Figure 2). These parts were glued together with a clean surface of titanium. In a series of “+B” samples, an additional technological spacer in the form of a 1 mm thick steel sheet was mounted between a 3D printed frame and a titanium plate. A TNT detonator was placed in the cavern. The space within the frame, which was 10 mm high, was filled with ammonal or emulsion explosive to completely cover the surface of the titanium. In the case of plastic-bonded explosives, a 5 mm frame was used. The connected system was carefully moved to the blasting bunker and was put on a steel base, which made it possible to increase the effects of the explosion by reflecting the shock wave and additionally compressing the titanium. A blasting cap was placed inside a TNT detonator and a detonation was initiated by an electric signal from a safe distance. After the experiment titanium parts were collected and cleaned. An example of the condition of the titanium plate after the last stage of the strengthening process is shown in Figure 2.

### 2.3. Strength Tests

Strength tests were carried out using a standard INSTRON SATEC 1200 kN testing machine in accordance with EN 10002-1:200. Test samples were taken from sheets deformed by the explosive method at an angle of 45° to the edge of the sheet, according to the schematic diagram shown in Figure 3. The angle of 45° was the angle of propagation of the detonation wave due to the exothermic reaction. Strength tests were carried out for all samples listed in Table 1.

### 2.4. Microhardness Tests

Microhardness tests were performed using a Vickers microhardness tester. The tests were carried out with a load of 1.96 N corresponding to 0.2 kG, in accordance with the PN-EN ISO 6507-1 standard. Measurements were made from the surface of the sheet at a distance of 0.3 mm through the entire section of the sample (8 mm) with a step of 0.3 mm.

### 2.5. Microstructural Tests

The samples for microstructural tests were cut with a band saw and prepared for testing by subjecting them to the process of grinding and polishing with a grinder and polisher using abrasive papers with a gradation of 600–1200, and a polishing cloth intended for titanium alloys and silica suspension. Microstructure studies were carried out using an AxioScope.A1 Zeiss (Oberkochen, Germany) light microscope. Test samples were prepared and etched using Kroll’s reagent. Microstructural tests were carried out for samples of the material in the initial state and after deformation using the PBX explosive.

### 2.6. Quantitative Analysis of the Microstructure

Microstructural analysis was performed on material samples previously prepared and etched to reveal the microstructure. The photos were formatted and prepared for analysis by graphic processing and making outlines of grain boundaries. The actual analysis process was performed using the Micrometer v1.0 program. On the basis of the data collected by the program, the parameters of the microstructure were determined: the equivalent diameter of grains (d), the average surface area of grains (A), and the relative surface of grains (Sv). In addition, based on the collected data, the standard deviation, variance, and the CV parameter, also known as the variability parameter, which is a measure of the standard deviation from the statistical mean, were determined.

### 2.7. Corrosion Tests

Corrosion tests were carried out using the Autolab PGSTAT3 (Metrohm Autolab B.V., Utrecht, The Netherlands) potentiostat. Electrochemical impedance spectroscopy and electrochemical polarization were used for corrosion tests. DC electrochemical tests were performed using the potentiodynamic method at a fixed polarization voltage of 10 mV/s. On the other hand, impedance tests were performed in the frequency range of 0.01 kHz-100 kHz, with an impulse with an amplitude of 10 mV. The materials selected for the corrosion tests were the samples whose strength properties changed the most compared to the initial state. The samples that were subjected to corrosion tests were tested in Ringer’s solution, the composition and characteristics of which are given in Table 3.

The results of the impedance measurements are presented in two different coordinate systems: Nyquist and Bode. The Nyquist method is a curve in the Re(Z)/Im(Z) system, in which the Bode plot shows two curves with a common axis of abscissa represented by log(f). The first curve in the ordinate axis is represented by the variable log|Z| representing the dependence of the impedance modulus and the second curve represented by the variable -Θ denoting the phase shift angle.

## 3. Results and Discussion

### 3.1. Strength Tests

The performed static tensile tests made it possible to estimate the impact of explosive strengthening on the basic strength parameters of the grade 2 titanium alloy. The mechanical properties of the base material have been presented for two various rolling directions: parallel (=) and perpendicular (+). The obtained results of tensile strength and elongation at break are presented below (Figure 4a,b).

Regardless of the explosive used and the method of loading the workpiece with the shock wave, an increase in tensile strength was recorded (Figure 4a). In the case of ammonal, which is the weakest explosive used, the increase is minimal and difficult to observe. A slightly greater strengthening (approx. 45 MPa) is observed in the titanium alloy strengthened by the detonation of the emulsion explosive. Reinforcement made with a plastic explosive allowed for a significant increase in strength, reaching a relatively tensile strength value of 727.3 ± 22.5 MPa. Analyzing the obtained results, it can be concluded that the ductility of the explosively treated alloy decreases with the increase in strength, which is manifested by a significant reduction in elongation at break, especially in the case of plastic explosives (PBX), approx. 6% (Figure 4b). It should be noted that the use of weaker materials (ammonal and emulsion explosives), despite relatively low strengthening, gives a positive effect in the form of an increase in the ductility of the tested alloy. This is an unexpected effect for the applied treatment and requires further research to clarify the exact mechanism of the increase in plasticity. The use of an intermediate layer in the form of a 1 mm steel plate between the explosive and the treated titanium alloy had a noticeable effect on the strength values obtained (Figure 4a). It should be mentioned here that the plate was not used as a hammer, but its only task was to eliminate the direct impact of the detonation products on the workpiece while weakening the pressure impulse. The smallest reduction in strengthening in relation to the system with a directly located explosive can be seen in the case of a plastic explosive for which the strengthening obtained is over 50 MPa lower and the strength value obtained is 675.2 ± 17.3 MPa (Figure 4a). At the same time, it is noteworthy that the use of an intermediate layer significantly improved the ductility of the treated plates (Figure 4b) and, even in the case of PBX, an elongation value exceeding 25% was achieved. The reason for this state of affairs should be seen in the reduction in the pressure impulse, which entails a correspondingly smaller defect in the structure.

In the case of the direct location of the PBX on the workpiece, due to the high-pressure impulse, the main role in strengthening is most likely played by phase transformation products, significantly reducing the ductility of the alloy. This case will be the subject of detailed microstructural studies in the next publication. Considering the impact of the conducted explosive strengthening processes on the increase in the strength of a grade 2 titanium alloy, one can refer to the detonation velocity of the material used (Figure 5).

Analyzing the above graph, it can be concluded that within the range of the detonation velocities used, it is possible to obtain a strengthening of grade 2 titanium up to a value of approx. 40% (Figure 5). The use of an intermediate steel plate allows for more effective hardening in the case of materials with detonation velocities up to approx. 6000 m/s. Above this value, less effective hardening is observed (Figure 5), but it should be noted that the alloy subjected to this treatment will be characterized by very good ductility (Figure 4b).

### 3.2. Microhardness Tests

The tests of microhardness in the Vickers scale were carried out on samples in the delivery state and those which were strengthened by the explosive method. The samples were tested using a microhardness tester to determine the hardness profile on the cross-section of the samples. The tests were carried out from the surface in direct contact with the explosive to the base of the tested material. The obtained microhardness distributions and maximum strengthening are shown in Figure 6a–c.

Figure 6a,b show the courses of microhardness profiles for all tested samples in relation to the initial state. Figure 6c shows a summary of the maximum hardness values obtained for the tested samples. It can be observed that the highest hardness and the widest strengthening zone are characteristic of the material strengthened with PBX and PBX +B explosives. The use of ammonal and 3% emulsion did not significantly change the microhardness of the sheets over the entire section. The lower hardness of the sample strengthened with ammonal in relation to the initial state is within the range of statistical error. The highest increase in hardness for all strengthened sheets was observed at the surface of the sheet which had direct contact with the detonating explosive, and thus was subjected to a strong shock wave. The shock wave causes a strong defect in the material; hence, the microhardness changes will be most visible in the area close to the surface it affected. The depth of the fortified zone varies depending on the explosive used. In the case of PBX, the pronounced impact of blasting can be estimated at about 2.5–3 mm from the hardened surface. After exceeding this distance, the hardness values in the PBX-hardened material are close to the hardness in the initial state and are within the statistical error. However, in the case of the PBX + B sample, a slightly lower maximum hardness value is observed in relation to the PBX sample. The difference in hardness for the first five measurement points is about 10 HV0.2.

The microhardness profile presented in Figure 6a for samples strengthened by the explosive method with ammonal and 3% emulsion, in relation to the initial state, which significantly differs from the previously presented microhardness profile for materials from the PBX group. In the case of samples strengthened with ammonal and 3% emulsion presented in the graph above, the differences from the initial state and the effect obtained after the explosive hardening process are small. The ammonal-strengthened titanium plate shows a slightly increased hardness at the very surface that was exposed to the shock wave. However, the maximum obtained hardness of the sample strengthened with ammonal relative to the starting material is the same, while for the sample strengthened with 3% emulsion, the increase is 29 HV0.2. For a distance from the surface not exceeding 1 mm, the strengthening effect is visible, while later the hardness is close to the hardness of the initial state. The use of an additional sheet during the strengthening process slightly increases the maximum hardness for the ammonium-reinforced sample, while for the samples strengthened with emulsion and PBX, a decrease in the maximum hardness value was observed in relation to the base material (Figure 6b). The difference in the degree of strengthening of the material by using an explosive with a high detonation velocity (PBX) and materials with a lower value of detonation velocity (ammonal, emulsion 3%) is, therefore, significant. The use of an explosive with a higher detonation rate is associated with the generation of a much higher pressure, which results in a more intensive strengthening of the tested material.

### 3.3. Microstructure—Initial State

The output material exhibits a grain structure characteristic of grade 2 titanium after hot rolling. The microstructure of the initial state is characterized by a typical, single-phase α type structure, where the grains are uniformly distributed and equixially shaped (Figure 7). These grains do not have a privileged orientation. 

### 3.4. Microstructure—Explosively Hardened Titanium with PBX

Carrying out the process of hardening by blasting has a significant impact on the microstructure of the processed material in the near-surface area. Strengthening of titanium with the PBX explosive significantly changed the nature of the microstructure of the deformed material in relation to the initial state provided by the manufacturer. The appearance of martensitic titanium in the material at a thickness of about 100 μm from the sheet surface was observed as a result of the impact of the shock wave on the material (Figure 8). Twin structures were observed in the rest of the material. They were created by introducing plastic deformation and shearing the crystal planes, which are responsible for the formation of twins.

### 3.5. Quantitative Analysis of the Microstructure

After performing the microstructure tests, a quantitative analysis of the microstructure of the tested samples was carried out in terms of grain size distribution and their equivalent diameter. Two representative samples were selected for testing. Grade 2 in the original state as delivered from the manufacturer and the material after strengthening with PBX material were selected, due to the greatest change in microstructure and properties relative to the original state. In the case of the state after explosive deformation, tests were performed using the Micrometer program, thanks to which the characteristics of the microstructure distribution in each state of the material were determined. The tests were carried out in a cross-section from the surface of the sample to its base, including the entire characteristics of the microstructure. The test results are presented in Table 4, taking into account statistical parameters such as variance, standard deviation, and CV parameter—coefficient of variation. 

The data presented in the table show differences in the average grain size of individual samples, which suggest changes in the microstructure as a result of the explosive hardening process. Relative to the initial state, the process of strengthening titanium with the PBX explosive caused a change in the average grain size. Relative to the initial state, the difference was 5–5.5 µm for the equivalent diameter of the grains. It is worth paying attention to the parameters of the standard deviation, which also inform about the uncertainty of the measurements. This indicates the existence of a grain size dispersion, which directly affects the nature of the result. Considering the standard deviation, there is almost no difference between these two samples, which can be explained by the fact that the measurements have been taken in the mid-thickness of the hardened plate, where the impact of a shock wave was relatively low. In addition, the CV parameter that was determined for each sample is also very similar for each sample. 

### 3.6. Corrosion Tests—Electrochemical Impedance Spectroscopy

In order to study the corrosion characteristics of the material, tests were performed using electrochemical impedance spectroscopy and direct current tests. The activity was aimed at determining the behavior of the material before and after the strengthening process in conditions simulating the environment of physiological fluids. The test was carried out at 37 °C in Ringer’s solution, and two representative samples were selected for testing. Grade 2 in the original state, as delivered from the manufacturer and the material after strengthening with PBX material, was selected due to the greatest change in microstructure and properties relative to the original state. The obtained test results were presented in the form of Nyquist and Bode plots separately for each tested material. Figure 9 and Figure 10 show the results of the tests in the form of Nyquist and Bode plots obtained after the analysis using the electrochemical impedance spectroscopy method. In the case in the Nyquist diagram, a simple course of the loop can be observed, which is not characterized by the presence of any additional elements, apart from the main single course of the capacitive loop; it means the occurrence of the corrosion processes is in accordance with the mechanism of charge exchange between the outer layer of the metal and the electrolyte. In the case of the Bode plot, a course characterizing the occurrence of corrosion according to this mechanism can also be observed. The Bode plot also shows a wide range of phase angle values that are close to the maximum value. This indicates a reduced occurrence of corrosion processes in this area of the tested frequency [21].

Figure 11 and Figure 12 show the results for a sample subjected to explosive strengthening with PBX material. In this sample, the Nyquist plot is similar to the baseline. A slight difference is observable for the measurement values that characterize both graphs due to the flatter course of the sample after explosive strengthening. In addition, the Bode diagram for the hardened sample is characterized by a lower range of occurrence of the phase angle value oscillating in the range of maximum values, and the phase angle itself is located lower than in the case of the initial sample [22].

In the diagram shown in Figure 13, one can see the differences in individual materials and their behavior in a corrosive environment. The sample in the initial state is characterized by the largest diameter. The sample subjected to explosive hardening has the smallest diameter, which indicates the impact of strengthening on the course of corrosion phenomena.

Additional characteristics are presented in the diagrams of the module |Z| and phase angle for collective Bode plots in Figure 14 and Figure 15. Analyzing the results, slight differences were found, especially in the case of the plot showing the value of |Z| as a function of frequency. However, for the value of the phase angle, a change can be observed in the form of the previously mentioned width of the range of occurrence of values close to the maximum phase angle. This indicates the characteristics of the intensity of the processes that occur between the metal surface and the electrolyte. In the case of the samples presented above, it can be seen that, according to previous observations and information presented in the Nyquist plot, the sequence and characteristics of the samples are similar. The sample of the starting material has the widest range, and the sample after strengthening with PBX material has the narrowest range [21,22,23,24].

### 3.7. Corrosion Tests—DC Electrochemical Research

In the further part of the research, polarization curves were determined, thanks to which it was possible to determine the corrosion potential and corrosion current density and to determine the characteristics of the material’s behavior at a changing electrochemical potential. The collective graph of the determined polarization curves is presented in Figure 16, and the values of the determined numerical data are presented in Table 5.

The course of polarization curves for the tested samples (starting material and PBX) is very similar and is typical for the passivating material. The part of the curve characterized as cathodic polarization and the part of the active state is almost identical for all samples and the course of the curve is similar. The transition to the passive state is almost instantaneous. The fluctuations in the graph in Figure 16 result from the periodic cycle of switching the heating lamp on and off. The values determined by DC electrochemical tests can also be described as approximate. The current densities are minimal and do not differ significantly, while the corrosion rates per year are negligible and, according to the characteristics presented in the impedance tests, the highest speed and intensity of corrosion processes are characteristic of explosively strengthened samples, while the smallest sample is in the initial state. However, the change in parameters is negligible and does not exceed the corrosion rate of 1 µm per year [25].

## 4. Summary

The tests carried out allowed us to identify and characterize the material in the form of grade 2 titanium subjected to explosive strengthening with selected explosives in the form of 3% emulsion, ammonal, and PBX. The explosive hardening process affected both the mechanical characteristics and the microstructure of the material. After carrying out the strengthening processes, as a result of the impact on the material with high pressure and stress through the detonation wave, the microstructure changes. The results of microstructural tests presented in the paper allow us to observe the resulting changes. In the case of the starting material, the microstructure is characterized by grains of various morphology and size. There is no pronounced texture despite the sheet rolling process. Another effect was the occurrence of martensitic transformation, which was also associated with the action of the shock wave. In the case of hardened PBX samples, this change was most noticeable. Microstructural analysis showed changes in grain size distribution. In relation to the initial state, the greatest fragmentation on the cross-section of the sample is observed for the PBX-reinforced material. The performed mechanical tests made it possible to determine the strength properties of the material after the strengthening process. In the case of microhardness analysis, the greatest changes can be seen in the case of measurements made in the vicinity of the deformed surface. For samples deformed with strong explosives, the change was the most visible and hardness values reached the highest values, while in the case of weaker materials, the change was small. The material characterized by the highest strength was the deformed PBX material. Conducting corrosion tests allowed us to determine the impact of deformation using the explosive method on the corrosion resistance of the material. Impedance tests have shown that the impact of the explosion on the material causes a decrease in the range in which the material is more resistant to corrosion. Both the determination of the polarization curves and the corrosion rate values confirm that the impact of the shock wave decreased the corrosion resistance of the material. However, the change in corrosion rate is less than 0.1 µm/year, which does not significantly affect the overall corrosion resistance of the material. The impact on the reduction in corrosion resistance is probably due to the change in the nature of the surface of the material.

## Figures and Tables

**Figure 1 materials-16-00847-f001:**
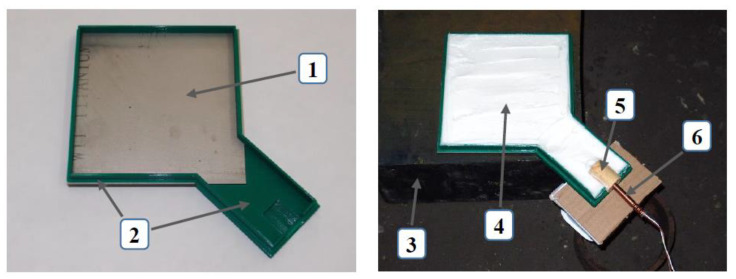
Technological system for strengthening titanium using the explosive method. 1—titanium plate, 2—elements obtained by additive techniques from PLA, 3—steel base, 4—explosive, 5—TNT detonator, and 6—blasting cap.

**Figure 2 materials-16-00847-f002:**
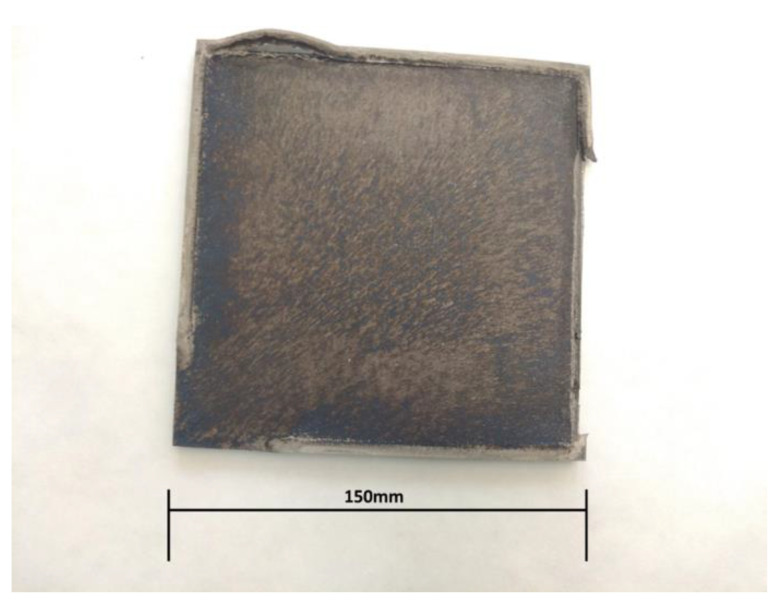
Titanium sheet after explosive hardening.

**Figure 3 materials-16-00847-f003:**
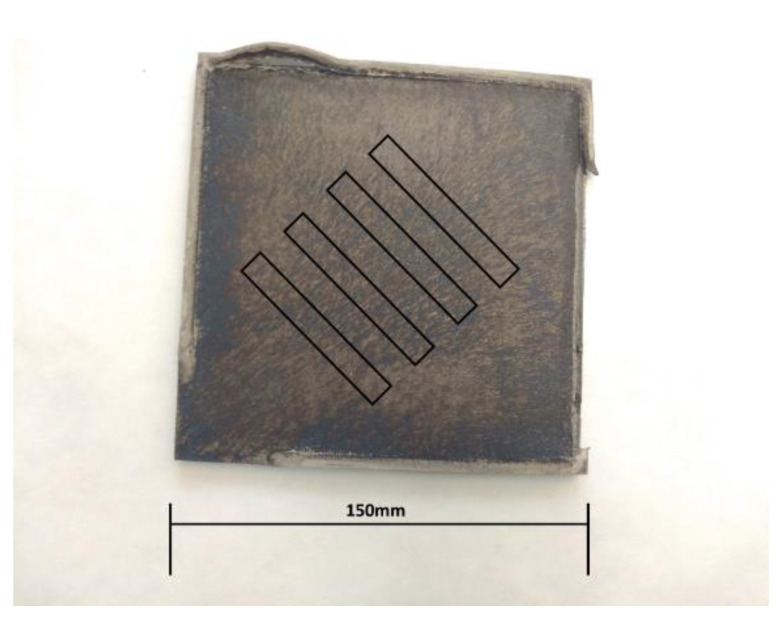
Diagram of the method of taking samples from sheet metal after deformation using the explosive method. Samples marked with a black outline are included at an angle of 45° to the edge of the bottom sheet.

**Figure 4 materials-16-00847-f004:**
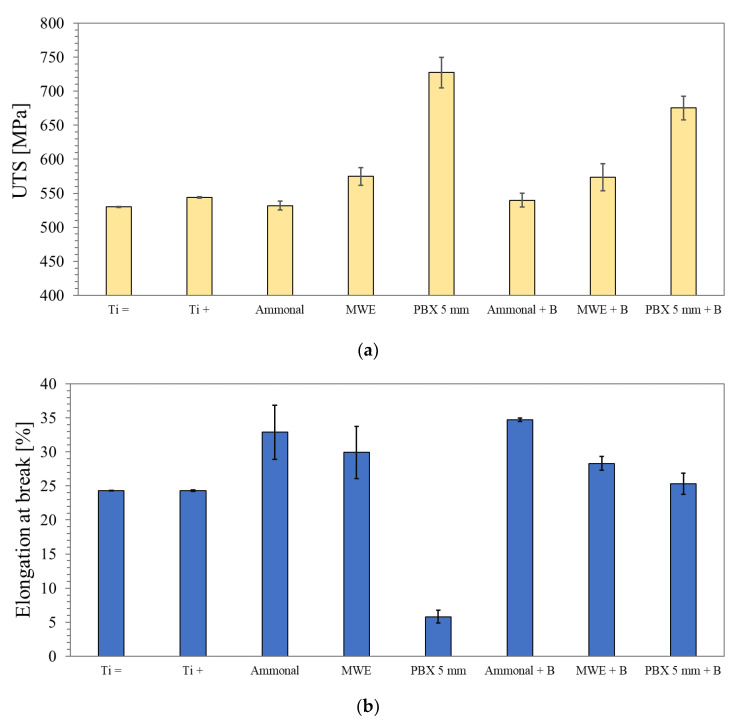
The obtained results of the tensile strength (**a**) and elongation at break (**b**) for different explosives and the use of an intermediate steel plate (+B).

**Figure 5 materials-16-00847-f005:**
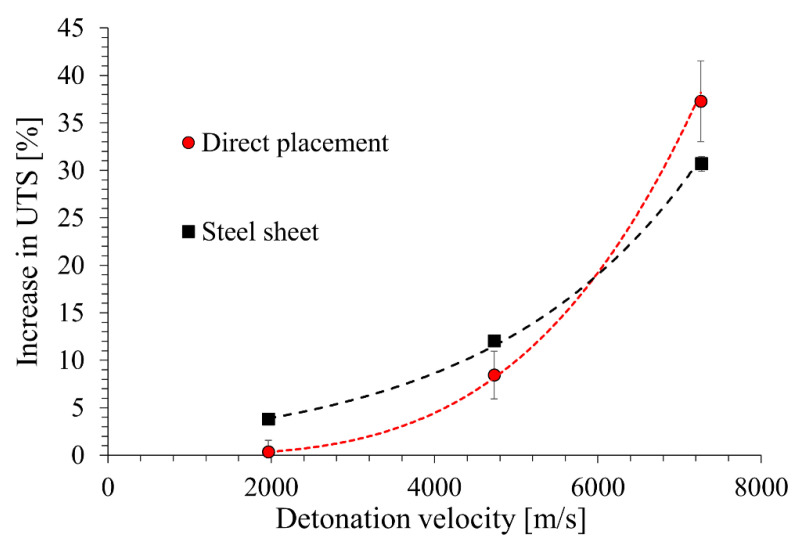
Increase in tensile strength depending on the speed of detonation of the explosive and the method of placing it on the processed plate: directly (direct placement) and through an indirect steel sheet.

**Figure 6 materials-16-00847-f006:**
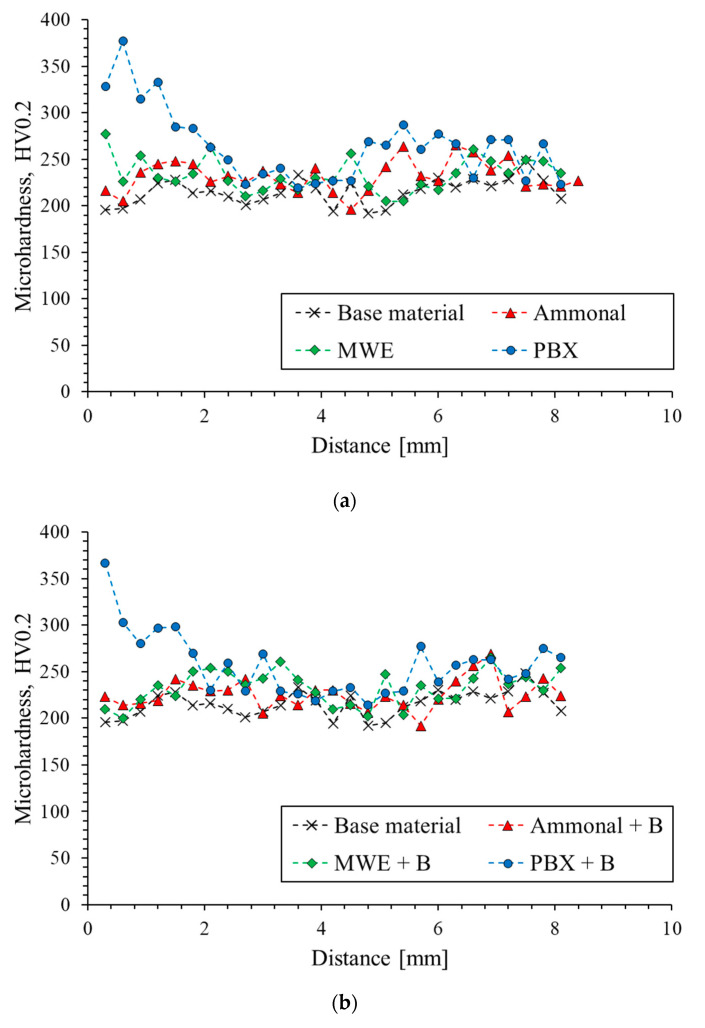
Distributions of microhardness obtained for samples without a steel sheet (**a**), with sa teel sheet (**b**), and the greatest strengthening obtained (**c**).

**Figure 7 materials-16-00847-f007:**
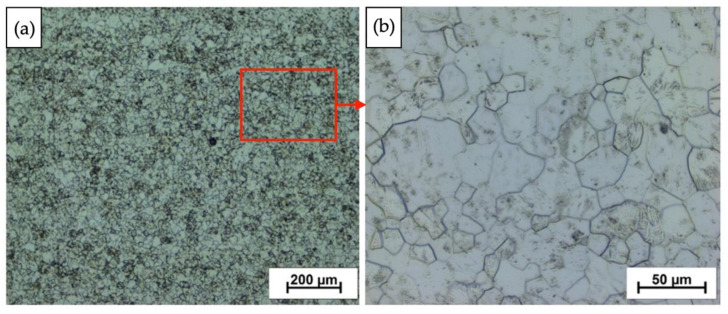
Initial condition after etching and before explosive hardening; grade 2 titanium with visible grain boundaries. (**a**) Macro image and (**b**) high magnification image (area indicated by the red rectangle).

**Figure 8 materials-16-00847-f008:**
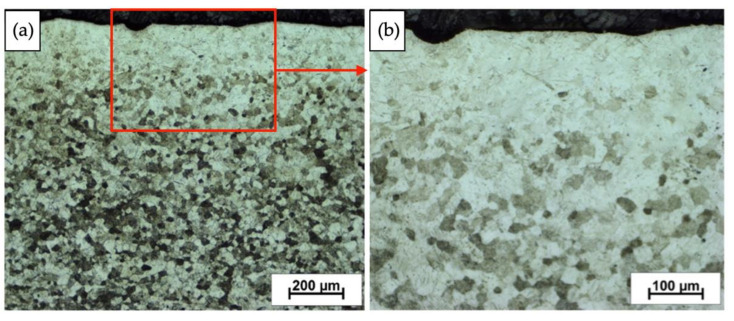
Fragment of a sample deformed with a PBX explosive with the surface area marked. Visible fragments with a coniferous structure. (**a**) Macro image and (**b**) high magnification image (indicated by the red rectangle).

**Figure 9 materials-16-00847-f009:**
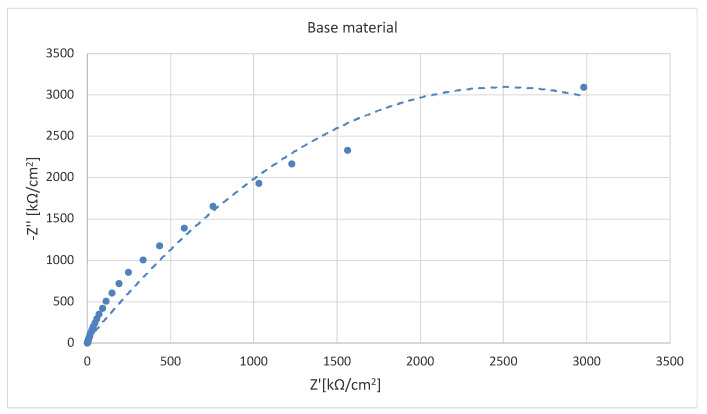
Nyquist plot for the sample in the initial state (supplied by the manufacturer).

**Figure 10 materials-16-00847-f010:**
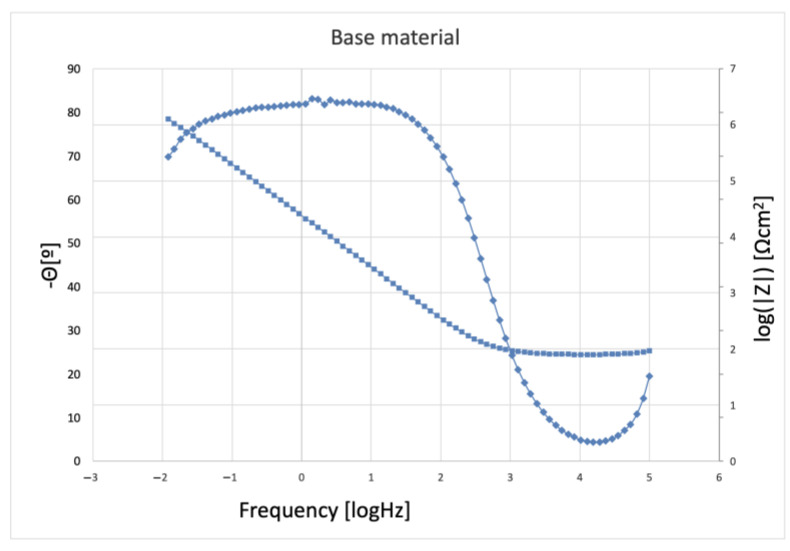
Bode plot for the initial state of the sample (obtained from the manufacturer).

**Figure 11 materials-16-00847-f011:**
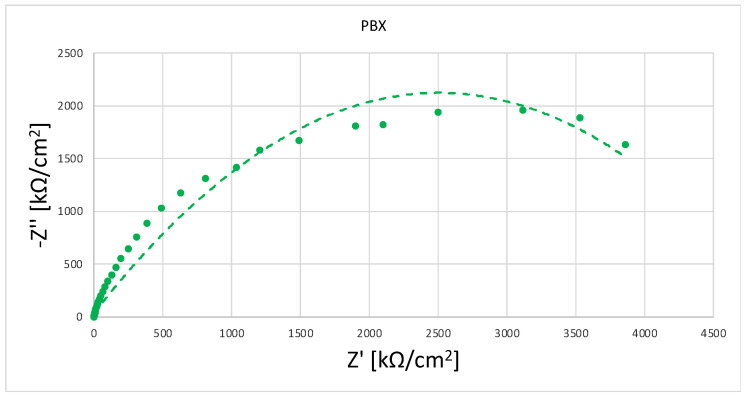
Nyquist plot for a sample strengthened with PBX material.

**Figure 12 materials-16-00847-f012:**
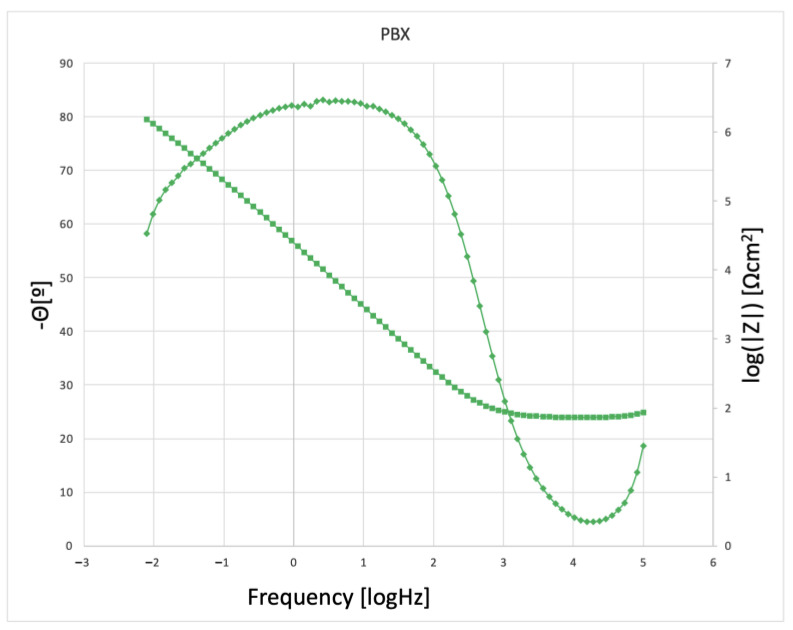
Bode plot for a PBX−enhanced sample.

**Figure 13 materials-16-00847-f013:**
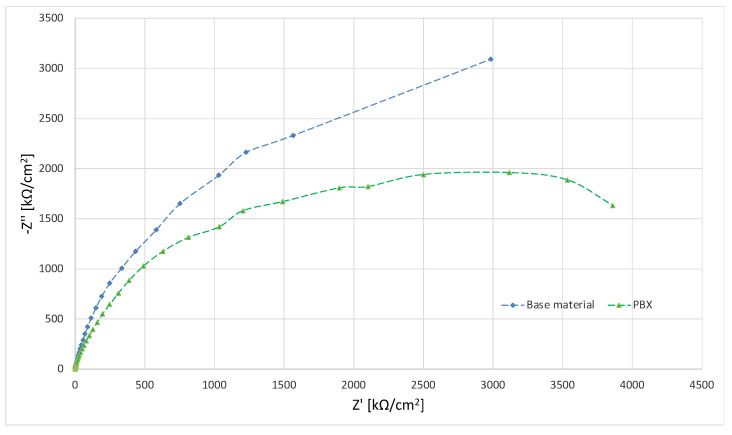
Nyquist summary plot for all tested samples.

**Figure 14 materials-16-00847-f014:**
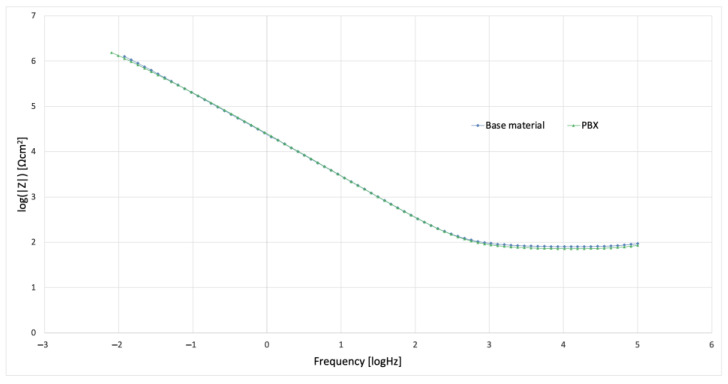
Bode summary plot for |Z| for the tested samples.

**Figure 15 materials-16-00847-f015:**
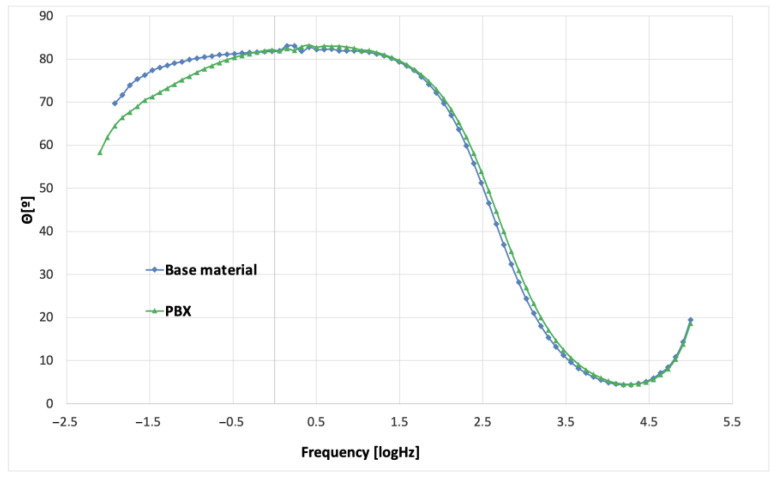
Bode summary plot for the phase angle values for the tested samples.

**Figure 16 materials-16-00847-f016:**
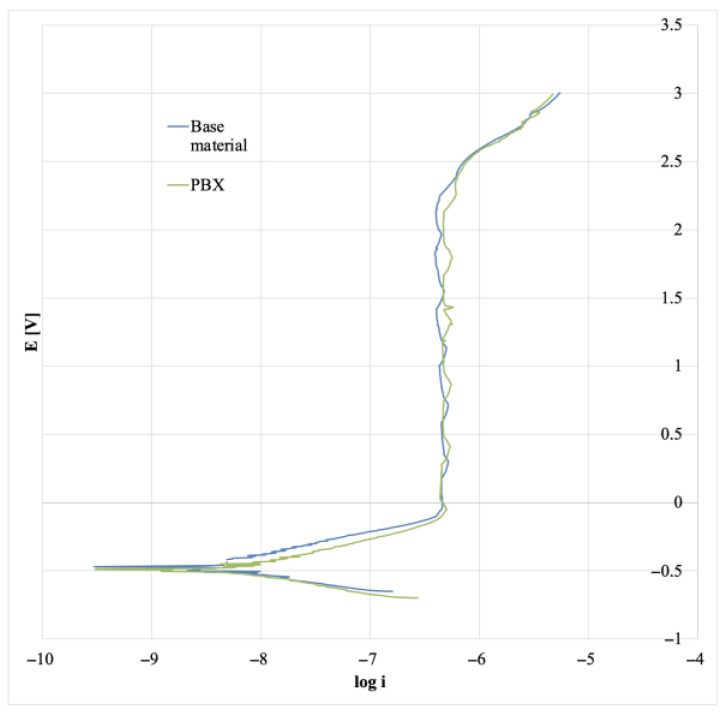
Summary plot for polarization curves for all samples.

**Table 1 materials-16-00847-t001:** Characteristics of the tested samples.

Designation	Description of the Samples
Base material	Sheet in the initial state provided by the manufacturer
Ammonal	Sheet deformed with an explosive—ammonal explosive
MWE	Sheet deformed with an explosive—emulsion explosive
PBX	Sheet deformed with an explosive—plastic bonded explosive
Ammonal + B	Sheet deformed with ammonal explosive with an additional technological spacer in the form of a 1 mm thick steel sheet
MWE + B	Sheet deformed with emulsion explosive with an additional technological spacer in the form of a 1 mm thick steel sheet
PBX + B	Sheet deformed with plastic bonded explosive with an additional technological spacer in the form of a 1 mm thick steel sheet

**Table 2 materials-16-00847-t002:** Characteristics of explosives.

Designation	Composition	Layer Thickness [mm]	Density [g/cm^3^]	Velocity of Detonation [m/s]
Ammonal	10.0% Al, 86.5% ammonium nitrate(V), and 3.5% dolomite	10	0.72	2000
Emulsion explosive	97% emulsion matrix and 3% glass microspheres	10	1.11	4700
Plastic bonded explosive	85% hexogen and 15% matrix (binder, plasticizer, and oil)	5	1.48	7300

**Table 3 materials-16-00847-t003:** Ringer’s solution was used for corrosion tests [20].

Ingredient	Content g/dm^3^
NaCl	6.0
KCl	0.075
CaCl2	0.1
NaHCO3	0.1

**Table 4 materials-16-00847-t004:** Results of the quantitative analysis of the microstructure.

State	Grain Quantity	Equivalent Diameter [µm]	Variance	Standard Deviation	CV
Base material	171	23.3	128.36	11.33	0.49
PBX	183	18.1	53.4	7.31	0.40

**Table 5 materials-16-00847-t005:** Values of potentials and density of corrosion currents for the tested samples and the value of the corrosion rate given in mm/year.

State	U [mV]	I [nA/cm^2^]	CR [mm/Year]
Baseline	−469	4.6	0.00008
PBX	−490	5.5	0.0001

## Data Availability

Not applicable.

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
