# Peer review of "Research on Explosive Hardening of Titanium Grade 2"

_materials, 2023, doi:10.3390/ma16020847_

Round 1

Reviewer 1 Report

The aim of this work was to check the possibility of increasing the basic strength parameters of grade 2 titanium alloy by explosive treatment, in particular, plastic bonded explosive is used. Another potential solution is the use of technically pure titanium and subjecting it to appropriate treatment to give it the highest possible strength parameters, similar to the strength of the Ti6Al4V alloy. However, the change in corrosion rate is less, anti-corrosion research can be deleted, for example, 3.7. Corrosion tests.

Reviewer 2 Report

The article is dedicated to the interesting and original theme but needs to be significantly reworked, experimental details should be specified and the obtained results are to be analyzed more broadly.

Necessary and recommended additions and corrections are the following:

1. Describe in detail the samples installation pattern. Was there anything on the top of the titanium plate with the deposited explosive material? How did you fix the whole assembling?

2. Explain the scheme for cutting samples for mechanical tests. Why are there no samples cut from the edges?

3. Does surface layer of the titanium plate warm up after explosion? Does titanium oxidation occur? Does it have any effect on hardness and strength of the surface layer?

4. Authors’ statement causes amazement that there is no texture in titanium sheet after hot rolling. It cannot be so. Equiaxed grains cannot indicate textureless material. Moreover, there are no data and no images of the structure in all geometric directions of the sheet. Please, explain this statement.

5. The quality of metallographic analysis should be greatly improved. The presented images do not show what is written in the article - martensite, etc. The magnifications in the images of the alloy in the initial and deformed material do not match (the images on the right in Figures 7 and 8). Why are etched grains not visible in the surface layer in the top of Figure 8? What is the structure there? It is necessary to analyze the structure at a higher magnification using SEM.

6. The motivation of this scientific work is to study the possibility of increasing the strength of titanium by impact processing as an alternative to severe plastic deformation, in particular, when obtaining strong medical implants, for example, hip prostheses. A conclusion based on the results of your work is desirable. What are the prospects for your technology? Is it possible to strengthen bulk samples, workpieces for titanium hip prostheses in this way? Such products have a final complex shape. Will the cold-hardened layer be removed after machining or can a titanium workpiece of complex shape be subjected to impact processing? How then to achieve the uniformity of explosives distribution and a uniform impact?

Reviewer 3 Report

Hereinafter some comments to improve its quality:

1.    The abbreviation PBX is introduced for the first time in the abstract. Please, define this explosive material before using its abbreviation.

2.    Some typos and grammatical errors are found in the article. For instance, in page 7 line 198, PBX instead of MW. Please check the article throughout. Besides, in page 6, authors used “elongation TO break” in the text and “elongation AT break” in the y-axis in Fig.4b. Please use one designation.

3.    In the x-axis of Figure 4.a, the authors mentioned Ti = and Ti +. What is the difference between these to designations?

4.    In page 7, the authors stated, “The use of an intermediate layer in the form of a 1 mm steel plate between the explosive and the treated titanium alloy had a noticeable effect on the strength values obtained (Fig. 4a)”. However, it is obviously noticed from Fig.4a that only the use of PBX as explosive material induced a significant change in strength. The same reasoning can be applied to Fig.4b. Authors should explain why the use of PBX affects the strengthening of the titanium alloy.

5.    In Fig.5, the tendency of the two curves changes at approximately 6000 m/s. Can authors explain the reason of this phenomenon?

6.    Please distinguish the two micrographs in Figures 7 and 8 by inserting labels such as “a” and “b” and modify the Figures captions consequently. 

7.    The authors mentioned in page 10 the formation of martensite accompanied by the apparition of twins. Please explain the mechanism of this phase transformation; was it a deformation induced martensite or does the martensite form during fast cooling from elevated temperature (in case the explosion generated local high temperatures)?

8.    An investigation using a SEM microscope would allowed a better characterization of the different microstructural evolutions that the alloy underwent.

9.    Considering the standard deviations in Table 4, the difference between the grain sizes of the base material and PBX is not really significant.

10.     Due to the microstructural changes induced during the hardening by explosion with the apparition of martensite, the mechanical properties of the alloy will depend in the morphology, distribution and mechanism of formation of martensite rather than in the initial alpha phase grain size.

Round 2

Reviewer 2 Report

 I agree to publish the manuscript in Materials at the current version. 

Author Response

The authors would like to thank the Reviewer for his committed time in reviewing this paper and for his constructive comments.

Reviewer 3 Report

9.    Considering the standard deviations in Table 4, the difference between the grain sizes of the base material and PBX is not really significant.

The reason for this is that the measurements have been taken in the mid-thickness of the hardened plate, where the impact of a shock wave was relatively low.

The comment was about the author's discussion of this result. In fact, they consider that there is a difference in grain size between the BM and the PBX. Whereas, considering the standard deviation, there is almost no difference.

Author Response

We fully agree with Reviewer's comment and we have improved the discussion on this subject.